# Assessment of Climate Biases in OpenIFS Version 43R3 across Model Horizontal Resolutions and Time Steps

Abhishek Savita[1], Joakim Kjellsson[1,2], Robin Pilch Kedzierski[3,1], Mojib Latif[1,2], Tabea Rahm[1,2], Sebastian Wahl[1] and Wonsun Park[4,5]

[1]GEOMAR Helmholtz Centre for Ocean Research Kiel, Kiel, Germany
[2]Faculty of Mathematics and Natural Sciences, Christian Albrechts University of Kiel, Kiel, Germany
[3]Meteorological Institute, Universität Hamburg, Hamburg, Germany
[4]Center for Climate Physics, Institute for Basic Science (IBS), Busan, Republic of Korea
[5]Department of Climate System, Pusan National University, Busan, Republic of Korea

*Correspondence to*: Abhishek Savita (asavita@geomar.de)

**Abstract.**

We examine the impact of horizontal resolution and model time step on climate of the OpenIFS version 43R3 atmosphere general circulation model. A series of simulations for the period 1979-2019 are conducted with various horizontal resolutions (i.e., ~100, ~50, and ~25 km) while maintaining the same time step (i.e., 15 minutes) and using different time steps (i.e., 60, 30 and 15 minutes) at 100 km horizontal resolution. We find that the surface zonal wind bias reduces significantly over certain regions such as the Southern Ocean, the Northern Hemisphere mid-latitudes, and in tropical and subtropical regions at high horizontal resolution (i.e., ~25 km). Similar improvement is evident too when using a coarse resolution model (~100 km) with a smaller time step (i.e., 30 and 15 minutes). We also find improvements in Rossby wave amplitude and phase speed as well as weather regime patterns when a smaller time step or higher horizontal resolution is used. The improvement in the wind bias when using the shorter time step is mostly due to an increase in shallow and mid-level convection that enhances vertical mixing in the lower troposphere. The enhanced mixing allows frictional effects to influence a deeper layer and reduces wind and wind speed throughout the troposphere. However, precipitation biases generally increase with higher horizontal resolution or smaller time step, whereas the surface-air temperature bias exhibits a small improvement over North America and the Eastern Eurasian continent. We argue that the bias improvement in the highest horizontal resolution (i.e., ~25 km) configuration benefits from a combination of both the enhanced horizontal resolution and the shorter time step. In summary, we demonstrate that by reducing the time step in the coarse resolution (~100 km) OpenIFS model, one can alleviate some climate biases at a lower cost than by increasing the horizontal resolution.

## 1. Introduction

In the last few decades, Atmospheric-Ocean General Circulation Model (AOGCM) simulations from the Coupled Model Intercomparison Project (CMIP) have been widely used to study the internal climate variability and the climate response to external forcing such as increasing atmospheric greenhouse gas concentrations causing global warming. These simulations, however, suffer from long-standing biases **(Bayr et al., 2018; Flato et al., 2014; Gates et al., 1999; Kim et al., 2014; Zhou et al., 2020)**, which lead to significant uncertainties in short-term and long-term climate projections and potential ecosystem impacts **(Athanasiadis et al., 2022; Couldrey et al., 2021; Meehl and Teng, 2014; Meng et al., 2022)**. These biases can arise from a variety of sources, including inaccurate representation of physical processes, poor initialization of model conditions, or inadequate representation of the Earth's topography and land cover.

Simulations using Atmospheric General Circulation Models (AGCMs) from the Atmosphere Model Intercomparison Project (AMIP), a part of CMIP, are used to study the internal variability of the atmosphere. The AGCMs are less complex than the AOGCMs as the former are constrained by observed Sea Surface Temperature (SST) and Sea Ice Concentration (SIC). Despite being constrained by the observations, the AGCMs also exhibit biases **(e.g., Gates et al., 1999)**, and some of these biases have persisted for over several phases of AMIP **(He and Zhou, 2014)**. The biases in AGCMs are largely due to the fact that many unresolved processes, such as atmospheric convection, precipitation, clouds, cloud-microphysical and aerosol processes, boundary layer processes, and interactions between the land surface and hydrologic processes, have to be included in a parameterized form in the coarse resolution model **(Ma et al., 2022)**. The treatment of unresolved gravity waves and the relatively large model time step also contribute to the biases in AGCMs **(Flato et al., 2014; Gates et al., 1999)**.

Recently, **Liu et al. (2022)** analyzed AOGCM simulations and reported that increasing the horizontal resolution of the ocean component one can reduce SST and precipitation biases in the equatorial Pacific, whereas increasing the horizontal resolution of the atmospheric component did not have the same effect. However, other studies found that a high-horizontal resolution atmosphere model better simulates the main features of tropical precipitation, tropical atmospheric circulation, and extra-tropical cyclones while increasing from 125 km to 40 km horizontal resolution with relatively small improvements for further enhanced horizontal resolution **(Branković and Gregory, 2001; Jung et al., 2012; Williamson et al., 1995).** Similarly, **Roberts et al. (2018)** found that there was not much improvement in the Integrated Forecasting System (IFS) from the European Centre for Medium-Range Weather Forecasting (ECMWF) when increasing horizontal resolution from 50 to 25 km.

**Jung et al. (2012)** and **Roberts et al. (2018)** demonstrated a time step sensitivity in the coarse and high horizontal resolution model simulations using the IFS model. **Jung et al. (2012)** found that the precipitation and wind biases were reduced at the

coarse horizontal resolution when shortening the model time step from 60 to 15 minutes. **Roberts et al. (2018)** did not find such a significant improvement when reducing the model time step from 20 to 15 minutes in their high-resolution (~25 km) configuration. However, both studies did not investigate the model's sensitivity to changes in the model time step in detail. While the semi-implicit semi-Lagrangian scheme, as used in OpenIFS, is unconditionally stable and the time step can be chosen to be very long, a shorter time step generally leads to a decrease in truncation error in the finite differences and thus a more accurate representation of the model dynamics. The physics parameterisations, which are computed independently of each other in OpenIFS, also benefit from a shorter time step as it will allow the various parameterisations to be coupled at a higher frequency (**Beljaars et al., 2018**). However, model parameters for e.g., convection or diffusion may be tuned for a specific time step and shortening the time step can therefore, in some cases, increase model error. Hence, a shorter model time step is expected to reduce biases in model dynamics, e.g., winds, while the results for parameterised processes, e.g., precipitation, may be mixed.

In the research community, there is no standard definition for coarse horizontal resolution, as one study considered 200 km as a coarse resolution (~2°) configuration (**Branković and Gregory,** 2001), whereas another study considered 50 km (0.5°) as a coarse resolution (**Roberts et al., 2018**). Likewise, there is no unique rule for setting the model time step dependent on model resolution. Groups using either the IFS or OpenIFS model at horizontal resolutions of ~100 km have used a relatively long time step of 1 hour (**Hazeleger et al., 2012; Kjellsson et al., 2020; Streffing et al., 2022**) or  45 minutes (**Döscher et al., 2022**), while other groups using the ARPEGE-Climat with a similar dynamical core use 15 minutes (**Voldoire et al., 2019**). The model's horizontal resolution and time steps are rather chosen on what can be afforded computationally, and their relative contributions to biases in the model's climate are not well documented.

In this study, we systematically investigate the sensitivity of the OpenIFS model version 43r3v2 to the model time step and horizontal resolution. We mostly focus on the surface zonal winds since they play a crucial role for the ocean circulation in the AOGCMs. We also study the representation of the synoptic-scale variability such as Rossby waves and weather regimes. The paper is structured as follows: section 2 describes the model, experimental design, data and methodology; section 3 describes the results and section 4 summarizes the conclusions of this work.

**2. Model, Experimental design, Data and Methodology**

We conducted a series of experiments with the OpenIFS model. The OpenIFS model is derived from the Integrated Forecasting System at the European Centre for Medium-range Weather Forecasting (ECMWF-IFS) cycle 43 release 3 (43r3). The dynamical core is the same as ECMWF-IFS that uses a two-time-level semi-implicit time stepping with semi-Lagrangian advection (**Temperton et al., 2001**) on a reduced Gaussian grid with a hybrid-sigma vertical coordinate (**Simmons and Burridge, 1981**). Likewise, the OpenIFS uses the same model physics as the ECMWF-IFS (**cf. Forbes and Tompkins, 2011;**

**Hogan and Bozzo, 2018; Tiedtke, 1993)** but does not include the tangent-linear code or 4D-VAR capabilities. Our version,
OpenIFS, is similar to cy43r1 used in **Roberts et al., (2018)**, with the main difference being the new radiation scheme, ecRad
(**Hogan and Bozzo, 2018**), introduced in cy43r3.

Our study is partly motivated by evaluating the suitability of various OpenIFS configurations for coupled climate simulations
with FOCI-OpenIFS **(Kjellsson et al.**, 2020) with an atmosphere horizontal resolution higher than that of ECHAM6 Tq63/N48
(~200km) in FOCI (**Matthes et al., 2020**). Our choices thus fall on three different horizontal resolutions: a low-resolution
($T_{co}$95, ~100 km), a medium-resolution ($T_{co}$199, ~50 km), and a high resolution ($T_{co}$399, ~25 km). The Tco95 grid is the lowest
acceptable resolution since the supported lower-resolution grids, e.g., Tl95/N48 and Tq42/F32, are either similar to Tq63 in
ECHAM6 or coarser. The Tco399 grid was chosen as an upper limit of what is computationally feasible for AMIP integrations
and century-long coupled integrations given our computer resources. All the configurations share the same vertical L91 grid.
We did not modify any other model parameters when changing the model horizontal resolutions or model time steps, but we
note that some parameters such as launch momentum flux for non-orographic gravity waves scales with resolution in the
model. We performed 5 experiments in total (**Table 1**). For simplicity, we now refer OpenIFS as OIFS in the rest of the
sections. We note that exploring the effect of different time steps was only done for the lowest horizontal resolution (Tco95,
~100 km). We did not run similar sensitivity experiments for the high-resolution configuration (Tco399, ~25 km) for two
reasons. First, the high-resolution configuration is very computationally expensive. Second, it was deemed more important to
explore time step sensitivity at low resolution since this configuration (and other similar resolutions) is often used for coupled
climate simulations. The potential time-step sensitivity at high-resolution is discussed in the Discussion section.


The lower boundary conditions, i.e., SST and SIC, are taken the from Atmospheric Model Intercomparison Project (AMIP)
version 1.1.6 **(Eyring et al., 2016; Taylor et al., 2012)**, which are available as monthly means on a 1º×1º horizontal grid. The
external forcing is identical to that used in the CMIP6 AMIP simulations except for the aerosol and ozone concentrations,
which are taken from monthly mean climatology. SST and SIC are interpolated from monthly to daily frequency and from
1º×1º horizontal resolution to the OIFS horizontal grid using bilinear interpolation. All the simulations are run for the period
1979–2019. We extend the simulations beyond the AMIP protocol for 1979-2014 up to 2019 by using SST and SIC from the
ERA5 reanalysis and the Shared Socioeconomic Pathways 5 (SSP5-8.5) emission scenario. Ozone concentrations are taken
from monthly photochemical equilibrium state and aerosol concentrations from monthly CAMS climatology of 11 species.

Amplitude and phase speed of Rossby wave were computed by performing a Fourier decomposition analysis on 300 hPa daily
meridional winds. First, we interpolated both ERA5 and OIFS simulation datasets onto a 2.5º x 2.5º grid using bilinear
interpolation. We then applied the Fourier decomposition analysis to determine amplitude and position for each Rossby wave
number at each latitude as a function of time. Phase speed is computed as the difference in the daily position of each wave,
and stored at the midpoints in the time dimension. For consistency, wave amplitudes are interpolated to the midpoints in time
as well. Lastly, seasonal averages are computed from the daily data for the boreal and austral winter seasons over the time
period 1979–2019. In the case of phase speed, it is weighed by the corresponding daily (midpoint) amplitude squared when
computing the seasonal averages in order to account for the impact of higher-amplitude events. The results are presented in
wavenumber-latitude diagrams similar to previous studies **(e.g., Pilch Kedzierski et al., 2020; Wolf and Wirth, 2017).** Our
wavenumber-latitude analysis is not directly comparable to both studies mentioned above, because we did not apply any high-
pass filtering in time before the Fourier decomposition. While the previous literature had similar diagrams with varying
measures of wave amplitude, our detailed analysis of phase speed in such a manner is novel in literature to our knowledge and
a strong addition as a model performance diagnostic.

The Weather Regime Patterns (WRPs) were calculated using daily 500-hPa geopotential height (z500) anomalies over the
Euro-Atlantic region (30°–90 °N, 80 °W–40 °E) for the boreal winter season during the period 1979-2019. The daily z500
daily anomalies were computed by subtracting the daily climatology smoothed by a 20-day running mean from the raw z500
data. We calculated the first four Empirical Orthogonal Functions (EOFs) from the ERA5 dataset. In the next step, the OIFS-
simulated z500 anomalies were projected on the ERA5 EOFs to obtain Pseudo-Principal Components (Pseudo-PCs). We then
applied a K-means clustering algorithm to the individual model pseudo-PCs and observation PCs using four clusters. We chose
four clusters because these give the most of the significant clustering. Spatial WRPs are obtained by compositing over all daily
z500 anomalies for each regime. More information about the methodology can be found in **Fabiano et al. (2020),** section 3.1.
In order to evaluate the WRPs simulated by the OIFS across configurations more quantitatively, we have additionally estimated
the Pearson's Pattern Correlation Coefficient (PCC) between the WRPs identified in the model and ERA5.

We compare the climate of OIFS to observational and reanalysis datasets. Precipitation is validated against the Global
Precipitation Climatology Project **(GPCP; Huffman et al., 1997)**, and the surface air temperature (SAT) against the
CRUTEM4 **(Harris et al., 2014; Osborn and Jones, 2014).** We have used the ERA5 reanalysis **(Hersbach et al., 2020)** to
evaluate 10-meter surface wind as well as the zonal wind at 300 hPa for the Rossby wave analysis. We use z500 from ERA5
to validate the OIFS-simulated weather regimes. We also compare our results with the MERRA2 reanalysis **(Gelaro et al.,**
**2017**) and find similar results. Therefore, the comparison with MERRA2 is not shown. The bootstrapping (in total 2000
iterations) method is used to compute the 95% confidence interval for the RMSE and the WRPs correlation.
**3. Results**
**3.1.1 Global and regional surface bias and deriving processes**
The annual mean 10m zonal wind (surface wind hereafter) bias during the period 1979–2019 for the different OIFS
configurations is shown in **Fig. 1**. We find that the OIFS-LRA-1h configuration has a large surface wind bias over most of the
world ocean, with positive biases in the mid-latitudes (the Southern Ocean, North Atlantic and North Pacific) and negative
surface wind biases over the tropical oceans (Tropical Pacific, Tropical Indian and Atlantic Ocean) (**Fig. 1b**). Thus, the OIFS-
LRA-1h configuration simulates too strong surface westerly winds (and wind speed) over the mid-latitude oceans, which, if
coupled to an ocean model, may cause biases in upper-ocean mixing and oceanic uptake of heat and carbon.

The surface wind bias in the OIFS-HRA-15m configuration is reduced significantly (**Fig. 1f**) over most of the world ocean
compared to the OIFS-LRA-1h configuration (**Fig. 1b**), indicating that increasing the horizontal resolution from 100 km to 25
km and shortening the time step from 1h to 15-min improves the representation of the surface winds. The surface wind bias
also significantly reduces everywhere in the OIFS-MRA-15m configuration (**Fig. 1e**) compared to the OIFS-LRA-1h
configuration (**Fig. 1b**). The surface wind bias in OIFS-MRA-15m is larger than that in the OIFS-HRA-15m configuration but
smaller than that in the OIFS-LRA-1h configuration. Similar conclusions are obtained by performing Root Mean Square Error
(RMSE) analysis, which shows that the OIFS-HRA-15m configuration has the lowest annual and global mean RMSE of
surface wind while the OIFS-LRA-1h configuration has the highest RMSE (**Fig. 2a, black line**). Though we have found a
significant improvement in the wind bias in the OIFA-HRA-15m configuration, it is not clear yet whether the improvement is
due to the increased horizontal resolution or the shorter time step.

Surface wind bias is also reduced in both the OIFS-LRA-30m (**Fig. 1c**) and OIFS-LRA-15m (**Fig. 1d**) configurations compared
to the OIFS-LRA-1h configuration (**Fig. 1b**), and the bias improvement is mostly observed at the same places as in the OIFS-
HRA-15m configuration (**Fig. 1f**). The surface-wind bias improvement is similar in the OIFS-LRA-30m and OIFS-LRA-15
configurations, except over the North Pacific and Southern Ocean where the OIFS-LRA-15m configuration has a smaller wind
bias than the OIFS-LRA-30m configuration. However, we have not seen a large difference between the OIFS-LRA-30m and
OIFS-LRA-15 configurations in the global average RMSE analysis (**Fig. 2a**).

The surface-wind bias improvement in the OIFS-HRA-15m and OIFS-LRA-15m configurations not only exists in the annual
average but also in boreal winter (DJF) and summer (JJA) (**Fig. 2a blue and red lines**, respectively). Our results are consistent
with **Jung et al. (2012)**, as they found a reduction in wind bias in the tropical Pacific region when they shortened the time step
in their coarse resolution configuration. However, this study and the **Jung et al. (2012)** study are not consistent with that of
**Robert et al. (2020)** who did not find much time-step sensitivity. We speculate that in **Robert et al. (2020),** the reduction
from 20 to 15 minutes in their high horizontal resolution (25 km) may be too small. Alternatively, the 20-minute time step
could be the optimal time step for the 25 km configuration.

The surface wind bias in the OIFS-HRA-15m and OIFS-LRA-15m configurations looks similar in pattern, but they differ in
magnitude. The OIFS-HRA-15m configuration has a smaller bias in the North Pacific, Peru upwelling and Agulhas Bank
regions compared to the OIFS-LRA-15m configuration. We hypothesize that the reduction in surface-wind bias in the OIFS-
HRA-15m configuration (**Fig. 1f**) compared to the OIFS-LRA-1h configuration (**Fig. 1b**) is a combination of the enhanced
horizontal resolution and shorter time step. The improvement in the OIFS-HRA-15m configuration (**Fig. 1f**) compared to the
OIFS-LRA-15m configuration (**Fig. 1d**) is due to only the enhanced horizontal resolution as both configurations use the same
time step.

The zonal-wind bias improvement in the OIFS-LRA-15m is further explored using the online zonal wind tendencies from
OIFS which are split into dynamics and physics that includes turbulent diffusion, gravity-wave drag and convection:

$$du/dt = du/dt_{Dyn} + du/dt_{Turb} + du/dt_{Gwd} + du/dt_{Conv} \quad (1)$$


where $du/dt_{Dyn}$ is the sum of the tendencies from advection, pressure gradient and Coriolis force, $du/dt_{Turb}$ includes tendencies
from surface processes, vertical diffusion and orography drag, $du/dt_{Gwd}$ includes gravity-wave drag and non-orographic drag,
and $du/dt_{conv}$ is the tendency from convection. The individual tendencies on the right-hand side of equation (1) are referred to
as Dyn, Turb, Gwd and Conv, respectively. They were stored for each model level in the OIFS-LRA-1h and OIFS-LRA-15m
configurations. The lowest model level is at 10m height (assuming surface pressure of 1013hPa), so the 10m wind will behave
very similarly to the wind at level k=91.

The averaged zonal wind and zonal wind tendencies over the Southern Ocean ($40^{o}$ S – $60^{o}$ S and all longitude) in the OIFS-
LRA-1h and OIFS-LRA-15m configurations are shown in **Fig. 3a & b,** respectively. The zonal wind tendency (i.e., $du/dt$) in
both OIFS-LRA-15m and OIFS-LRA-1h configurations is very small (~-2 to 0.04 m/s$^{-1}$) compared to the other processes (**Fig.**
**3b, black lines**). Conv provides westward acceleration between the 700 and 900 hPa pressure levels and eastward acceleration
below, indicating a downward transport of westward momentum. Dyn acts to accelerate the flow eastward from 700 hPa and
below, likely via momentum advection, pressure-gradient and Coriolis forces, while Turb has the opposite effect, likely via
surface friction and vertical mixing processes. In the OIFS-LRA-15m configuration, we find a similar balance as in the OIFS-
LRA-1h, but the westward acceleration above and eastward acceleration below is enhanced by Conv, likely by increased
downward momentum transport, in agreement with the increased shallow and mid-level convection (**Fig. 3d**). The vertical
momentum mixing by shallow and mid-level convection reduces the vertical wind shear, making the westerly winds more
barotropic. As a result, the westerly winds weaken throughout the troposphere and even in the stratosphere (**Fig. 3a**). We note
similar changes in the Northern Hemisphere mid-latitudes, suggesting similar mechanisms are acting. Gwd has a negligible
role for the winds in the lower stratosphere and troposphere, and the Gwd term does not appear sensitive to model time step
(**Fig. 3b, orange lines**).

**Fig. 3c** shows the zonal average of the zonal wind tendencies at the lowest level of the model, as a function of the latitude. In
the OIFS-LRA-1h configuration, Conv and Dyn accelerate the surface westerly wind in the mid-latitudes (~$40^{o}$ N to ~$60^{o}$ N)
in both hemispheres, and these westerly winds are partly balanced by Turb (**Fig. 3c, solid lines**). Dyn has a larger contribution
to accelerating the surface westerly winds than Conv (**Fig. 3c, solid lines**). However, the Conv contribution is enhanced in the
OIFS-LRA-15m configuration, while the Dyn contribution reduces (**Fig. 3c, dashed lines**). We also find that the contribution
to slowing the westerly wind is reduced by Turb in the OIFS-LRA-15m configuration (**Fig. 3c, dashed lines**).

It is also noteworthy that the individual wind tendencies are significantly larger in the Southern Hemisphere (and Southern
Ocean) than in the Northern Hemisphere (**Fig. 3c**). The larger magnitudes of the tendencies over the Southern Ocean compared
to similar latitudes on the Northern Hemisphere is likely due to the Southern Hemisphere having fewer continents in
midlatitudes than the Northern Hemisphere and thus the surface is less rough and allows for stronger winds. In the low latitudes,
both Dyn and Conv contribute to accelerating the easterly winds, which is partly balanced by Turb in the OIFS-LRA-1h
configuration (**Fig. 3c, solid lines**). There are no discernible changes in Conv, Dyn or Turb from OIFS-LRA-1h to OIFS-LRA-
15m, indicating that the tropical surface winds are relatively insensitive to model time step (**Fig. 3c, dashed lines**).

In addition to surface wind, we also investigated the sensitivity of model time step and horizontal resolution for SAT and
precipitation. The RMSEs for SAT and precipitation are shown in Figure 2b and 2c, respectively. We find that the OIFS-HRA-
15m has the lowest SAT RMSE of all model experiments in both annual and seasonal means, although the RMSE difference
across the configurations is not significant (**Fig. 2b**). The reduced SAT RMSE in OIFS-HRA-15m configuration is primarily
due to the lowered SAT bias over North America and the eastern part of Russia. Compared to the OIFS-LRA-1h, the SAT
RMSE decreases with increased horizontal resolution (OIFS-HRA-15m and OIFS-MRA-15m), and there is no notable
improvement when shortened the time step (OIFS-LRA-30m and OIFS-LRA-15m) (**Fig 2b**).

We have computed the SAT and precipitation biases with a 3-point smoothing, i.e., approximately 3x3 degree spatial
smoothing, which eliminates the wiggles near steep topography arising from the Gibbs' phenomenon in the model spectral
fields. We find that smoothing the fields does not change the main result that precipitation biases increase with shorter time
step in Tco95 and then decreases somewhat with higher horizontal resolution. Hence, the wiggles are not the main source of
precipitation biases and their presence does not impact the findings of this study.

The OIFS-LRA-1h experiment exhibits the lowest precipitation RMSE of all experiments, with RMSE increasing with shorter
time step (OIFS-LRA-15m) and increased horizontal resolution (OIFS-HRA-15m) for both the annual and boreal winter means
(**Fig. 2c, black and blue lines**). The patterns of regional precipitation biases are similar across the configurations in the mid-
and high-latitudes, whereas the precipitation biases increase in the tropics at the high horizontal resolution or in the smaller
time step configuration (not shown). The results suggest that some of the cloud and/or convection parameters may be dependent
on resolution or time step and need retuning for each configuration.

### 3.1.2 Wind and temperature bias in upper-atmosphere

We examined the zonal mean u wind bias at different model levels, and it is shown in **Fig. 4**. We find that zonal mean u wind bias over the tropical region (40°S and 40°N) is positive and independent of model horizontal resolution and model time step (**Fig. 4b-f**). The OIFS-HRA-15m configuration has a relatively large negative bias in the Northern Hemisphere compared to the other configurations. The OIFS-LRA-15m and OIFS-HRA-15m zonal mean u wind bias is similar to that in **Robert et al. (2018).** However, the zonal mean u wind bias in the Southern Hemisphere is not consistent across horizontal resolution or model time step. The zonal mean u wind bias in midlatitudes (i.e., 70°S to 50°S) is positive and large in the OIFS-LRA-1h configuration and reduces throughout the pressure levels by shortening the model time step in the coarse resolution OpenIFS configuration (i.e., OIFS-LRA-30m and OIFS-LRA-15m). Whereas, the negative zonal wind bias south 70°S in the coarse resolution configuration is consistent across the different time steps (**Fig. 3b-d**). It is also interesting to note that both OIFS-MRA-15m and OIFS-HRA-15m configurations exhibit a negative bias over the Southern Ocean (SO) at most of the pressure levels, which is not seen in the standard OIFS-LRA-1h, nor in the OIFS-LRA-30m or OIFS-LRA-15m configurations. Overall, we conclude that by reducing the model time step in the coarse resolution configuration, we improve winds not only at the surface but also at higher model levels mostly over the SO. A similar conclusion does not hold for the OIFS-MRA-15m and OIFS-HRA-15m configurations, as both suffer from large negative bias over the SO.

We also examined the zonal mean temperature bias at different pressure levels. We find a cold bias (1.5 to 6 °C) in the troposphere and lower stratosphere, and a warm bias (1.5 to 6 °C) above the stratosphere across the configurations (Figure not shown). This indicates that OpenIFS simulations (independent of model time step and horizontal resolution) are colder than observations in the lower stratosphere and warmer above. The cold bias in the lower stratosphere is larger in the high resolution (i.e., OIFS-HRA-15m), and the warm bias above the stratosphere is smaller compared to the other configurations. Robert et al. (2018) noticed a similar zonal mean temperature bias and speculated that the zonal mean temperature bias is linked with the sensitivity of spurious mixing due to convection and diffusion.

### 3.2 Rossby wave analysis

**Fig. 5** shows the Rossby wave amplitude (gray and black contours) for ERA5 and the individual OIFS simulations for the boreal winter (**Fig. 5A, DJF, Northern Hemisphere; NH)**) and austral winter (**Fig. 5B, JJA, Southern Hemisphere; SH**). The color in **Fig. 5** denotes the wave amplitude bias relative to ERA5 (model – ERA5). We focus only on those wave numbers and latitudes that have the highest wave amplitude, because these waves explain most of the variability. The region where the wave amplitude is larger than 5 ms$^{-1}$ is termed "core region", which mostly covers the area that is occupied by the thick black contours in **Fig. 5**. In DJF (NH), at north of 70° N, the Rossby wave numbers k=1 and k=2 have the largest amplitude in ERA5 whereas at the mid-latitudes (30° N to 60° N), the wave numbers between about k=3 and k=9 have large amplitude with the largest amplitude amounting to 8 ms$^{-1}$ at about 40° N for the wave number k=6 (**Fig. 5Aa**). During JJA (SH), the wave

amplitude is located in a similar core region (**Fig. 5Ba)** as that in DJF (NH). The amplitude is largest south of 70° S for the
wave numbers k=1 and k=2 whereas at the mid-latitudes (45° S to 65° S), the wave numbers between about k=3 to 5 have large
amplitude with the largest amplitude amounting to 9 ms$^{-1}$ is found at 57.5° S for the wave number k=4 (**Fig. 5Ba**).

In DJF (NH) the OIFS-LRA-1h configuration exhibits a positive bias of ~1 ms$^{-1}$ in Rossby wave amplitude (i.e., the waves
amplitude bias in OIFS-LRA-1h is larger than the ERA-5) in the core region, in particular for wave numbers k=3-8 at latitudes
between 25° N to 55° N and a negative bias at latitudes between 60° N to 80° N for waver number 2 (**Fig. 5Af**). The wave
amplitude biases around the core region in OIFS-LRA-1h in the midlatitudes (20° N to 40° N) are small (~0.2) for the higher
wave numbers and get better with a shorter time step configuration (OIFS-LRA-15m).

The Rossby wave amplitude biases in the OIFS-HRA-15m configuration are strongly reduced compared to the OIFS-LRA-1h
configuration over the core region (**Fig. 5Ab and 5Af**). The Rossby wave amplitude bias reduction in the OIFS-MRA-15m
configuration is mostly similar to that in the OIFS-HRA-15m configuration except for the wave number k=7 at 45° N, where
the wave amplitude bias is larger in the OIFS-HRA-15m configuration (**Fig. 5Ab and 5Ac**). The OIFS-HRA-15 m and OIFS-
MRA-15m configurations also exhibit a positive bias for wave number 2 at high-latitudes 60° N to 80° N.  The OIFS-MRA-
15m configuration also show a negative bias for the wave number 3 at latitudes between 60° N to 65° N in the core region,
which is not present in the other configurations. The OIFS-HRA-15m and OIFS-MRA-15m configurations show similar bias
around the core region as in the OIFS-LRA-1h configuration, i.e., high resolution and OIFS-LRA-1h configurations
overestimate wave amplitudes for the higher wave numbers. The Rossby wave amplitude biases are progressively reduced
from the OIFS-LRA-1h configuration to the OIFS-LRA-30m and OIFS-LRA-15m configurations (**Fig. 5Ad-Af**), indicating a
sensitivity of model bias to the time step. The wave amplitude bias for wave number k=7 at 45° N exists in all the
configurations, and it is smaller in the OIFS-LRA-15m and OIFS-MRA-15m configuration than in the other configurations.
Overall, both OIFS-LRA-15m and OIFS-HRA-15m configurations are able to reproduce the observed Rossby-wave
amplitudes in DJF (NH) better than OIFS-LRA-1h.

In JJA (SH), the Rossby wave amplitude bias in the core region is smaller than in DJF (NH) for all the configurations (**Fig.**
**5A and 5B**). OIFS-LRA-1h exhibits a positive bias of ~0.5 ms$^{-1}$ in JJA (SH) for the wave number k=2 at latitude between ~50°
S and ~62.5° S and for wave numbers k=4 to 5 between 30° S and 40° S (**Fig. 5Bf**). The OIFS-LRA-30m configuration shows
a positive bias for the wave number k=2 to 5 at latitudes between 40° S and 70° S, which is larger than other configurations.
The OIFS-HRA-15m and OIFS-MRA-15m configurations exhibits a positive bias ~0.5 ms$^{-1}$ around the core region and latitude
50° S to 70°S, which does not exist in the other coarse resolution configurations (**Fig. 5Bb-Bf).** The Rossby wave amplitude
biases around the core region at the midlatitudes in the high-resolution simulations are consistent and large in the SH than the
NH (**Fig. 5Ab-c and 5Bb-c**).

We also analyze the phase speed of Rossby waves for ERA5 and across the OIFS' configurations for DJF (NH) and JJA (SH)
seasons (**Fig. 6**). In the ERA5 dataset (**Fig. 6Aa**), the Rossby wave phase speed is positive (i.e., eastward moving, solid contour)
for wave numbers greater than 2 (i.e., k>2) at most latitudes. The wave numbers k=1 to 2 have a positive wave phase speed
from the equator to 55º N and a negative wave phase speed (i.e., westward moving, dashed contours) between 60º N and 80º
N in DJF (NH) (**Fig. 6Aa**). The maximum phase speed is found at wave number k=8 at 40º N, while the minimum is found at
wave number k=1 at 60º N (**Fig. 6Aa**). In JJA (SH) (**Fig. 6Ba**), the wave phase speeds are mostly positive and large for all the
wave numbers and at each latitude, with the maximum phase speed is observed for the wave numbers between k=6 and k=8
and latitudes between 40º S and 60º S, and these waves are moving faster than that in DJF (NH).

The OIFS-LRA-1h configuration suffers from positive phase speed bias for wave numbers k=4 to 8 at latitudes between 42.5º
N and 60º N, i.e., waves move faster eastward than in ERA5, and the bias is larger than 1 ms$^{-1}$. The bias of ~1 ms$^{-1}$ for wave
number k = 6 to 8 at 40º N and 60º N is of particular concern as it is near the maximum wave amplitudes in DJF (**Fig. 6Af**).
In general, phase speed biases in the OIFS-LRA-1h configuration are strongly reduced as either horizontal resolution is
increased or time step is shortened (**Fig. 6Ab-5Af**). In JJA (SH), the OIFS-LRA-1h configuration exhibits a very large (between
~1.5-2 ms$^{-1}$) Rossby wave phase speed bias for most of the wave numbers, which is largest for the wave numbers k=2 to 8
between 15º S to 55º S (**Fig. 6Bf**). Large biases can be found between 15º S and 25º S (~1.5 ms$^{-1}$) for most of the wave
numbers, but the wave activity is low there (**Fig. 6Bf**). The large phase speed biases are strongly reduced in the OIFS-LRA-
30m and OIFS-LRA-15m configurations (**Fig. 6Bd-Bf**), indicating a strong sensitivity to the reduced biases in mean winds
and wind speeds (**Fig. 1**). Overall, the Rossby wave speed bias in the OIFS-HRA-15m configuration is smaller than in the
OIFS-LRA-1h configuration (**Fig. 6Bb and 6Bf**). However, we note that both the OIFS-MRA-15m and OIFS-HRA-15m
configurations exhibit negative biases south of 55º S for wave numbers k= 1 to 5, that is, the eastward moving waves are
slower than in the ERA5 (**Fig. 6Bb**).

The wave phase speed analysis reveals a clear improvement in the representation of the Rossby waves in the boreal winter
(i.e., NH) when increasing the horizontal resolution and shortening the model time step compared to OIFS-LRA-1h
configuration. In austral winter, however, the representation of Rossby wave amplitudes and phase speeds are the most realistic
in OIFS-LRA-15m configuration, with longer time steps introducing too fast phase speeds and higher horizontal resolution
introducing too slow phase speeds at wave number less than 6 (i.e., k<6).
**3.3 Weather regimes pattern**
We derive the four weather regimes patterns (WRPs) over NH in the Euro-Atlantic region from ERA5. The patterns resemble
the positive and negative phases of the North Atlantic Oscillation (NAO+ and NAO–, respectively), Scandinavian blocking
(Sc. Blocking), and the North Atlantic ridge (Atl. Ridge) pattern (**Fig. 7, bottom row**). These WRPs are consistent with the
previous findings **(Dawson et al., 2012; Fabiano et al., 2020; Fabiano et al., 2021)**.

The OIFS-HRA-15m configuration produces WRPs that are more visually similar to those in ERA-5 than does OIFS-LRA-1h
(**Fig. 7**), a result confirmed by the higher pattern correlation coefficient (PCC) between OIFS-HRA-15m and ERA5 compared
to the OIFS-LRA-1h and ERA-5 (**Fig. 7 and 8**). The PCCs for NAO+, NAO- and Sc. Blocking all exceed 0.8 in OIFS-HRA-
15m while OIFS-LRA-1h does not achieve PCC above 0.8 for any WRP (**Fig. 8**).

The OIFS-MRA-15m configuration shows smaller PCCs than both the OIFS-HRA-15m and OIFS-LRA-1h configurations
(**Fig. 8**), i.e., the improvement from OIFS-LRA-1h to OIFS-HRA-15m does not have a linear relationship with model
horizontal resolution or time step. Compared to other configurations and ERA5, OIFS-MRA-15m the z500 anomaly in the
NAO+ pattern is too elongated in the southwest-northeast direction, and an unrealistic negative z500 anomaly over the North
Atlantic appears in the Sc. Blocking regime (**Fig 7**). Furthermore, OIFS-MRA-15m shows an Atl. Ridge pattern with neither
the right structure nor amplitude.

There is an improvement in the representation of the NAO- regime in the OIFS-LRA-30m configuration over the OIFS-LRA-
1h configuration (**Fig. 7)** while the Sc. Blocking regime becomes worse due to the ridge shifting westward. These changes are
also reflected in the PCCs (**Fig 8**). Similarly, the OIFS-LRA-15m better represents NAO- and Atl. Ridge than OIFS-LRA-1h
while NAO+ and Sc. Blocking worsened. The westward shift of the Sc. Blocking is similar in OIFS-LRA-15m and OIFS-
LRA-30m, and the worse NAO+ is related to a northward shift of both the positive and negative z500 anomalies. We note that
all experiments use the same SST and sea-ice conditions and that OIFS-LRA-1h, 30m and 15m share the same horizontal
resolution, i.e., the changes from OIFS-LRA-1h to OIFS-LRA-15m are not due to SST biases or representation of orography.
There does not seem to be a clear improvement as time step is shortened, despite the reduction in mean state biases and Rossby-
wave amplitudes and phase speeds.

The PCC is greater than 0.8 for three out of four WRPs in the OIFS-HRA-15m configuration, hence we argue that the OIFS-
HRA-15m has the most realistic representation of the weather regimes pattern out of all experiments here. Large improvement
in OIFS-HRA-15m over the other configurations could be due to better resolved topography and land-sea contrasts.

**3.4 Discussion and Conclusions**
We have investigated the sensitivity of the climate biases in the OpenIFS atmosphere model to changes in horizontal resolution
and time step by analyzing AMIP simulations for the period 1979-2019 (**Table 1**). The strong positive surface zonal wind bias

over the Southern Ocean and Northern Hemisphere mid-latitudes and the negative bias in the tropical and subtropical regions have significantly improved in the high horizontal resolution configuration with a short time step (~25km, OIFS-HRA-15m). A similar improvement is observed at the coarse horizontal resolution version with a shorter time step (~100 km with 30 or 15-minutes). The zonal wind bias over the mid-latitudes in both hemispheres is reduced throughout the air column when a smaller time step is used in the coarse resolution version, and we find that the changes in the surface winds are largely due to enhanced shallow and mid-level convection which increases vertical momentum transport. Biases in the surface westerlies in midlatitudes are common in CMIP-class climate models (**Bracegirdle et al., 2020**) and a sensitivity to friction has been noted in idealized model studies (**Chen and Plumb, 2009**). We hypothesize that the enhanced shallow and mid-level convection with a shorter model time step and/or increased horizontal resolution deepened the layer over which friction acts in the lower troposphere so that the frictional effects on the barotropic jet increased, leading to a poleward shift in the jet and reduced biases in zonal wind.

We also find a notable improvement in the representation of the Rossby wave amplitude and phase speed with increased horizontal resolution and shorter time step at least for the waves accounting for most variability in both austral and boreal winter seasons. The reduced zonal wind throughout the troposphere with a shorter time step (**Fig. 3**) would decrease the eastward phase speed of Rossby waves, which may explain part of the reduced phase speeds (**Fig. 6**) and reduced biases. However, changes in air-sea interactions or eddy-mean flow interactions may also play a role. In particular, we note that a very large reduction in phase speed biases in austral winter in OIFS-LRA-15m compared to OIFS-LRA-1h were concurrent with very large reduction in zonal surface wind biases.

The weather regime patterns are also more realistic in the high horizontal resolution and short time step configuration OIFS-HRA-15m than OIFS-LRA-1h, but we note that there is no consistent improvement from OIFS-LRA-1h to OIFS-HRA-15m as either horizontal resolution is increased or time step is shortened. For example, both OIFS-MRA-15m and OIFS-LRA-15m are worse than OIFS-LRA-1h. The improvements in the weather regime patterns and Rossby wave amplitude and speed could very well be related to each other as e.g. variations in Rossby wave breaking have been linked to the onset of NAO phases (**Strong and Magnusdottir, 2008**) but this would require further and more targeted analysis. The overall good representation of weather regimes in OIFS-LRA-1h compared to simulations with shorter time steps (OIFS-LRA-30m, OIFS-LRA-15m) may be due to compensation of errors. For example, it is possible that improving the wave amplitudes and phase speeds in OIFS-LRA-30m compared to OIFS-LRA-1h exposes the effect of a biases caused by both the coarse resolution in both configurations, e.g., weak interactions with topography, leading to an overall worse representation of weather regimes.

We found a gradual reduction in SAT biases in OpenIFS with increased resolution or shorter time steps. The improvements were largely driven by improvements over North America and eastern Russia. **Roberts et al. (2018)** noted similar SAT biases and linked them to surface albedo, which is thus likely the cause here as well. The improvement with increased resolution and/or shorter time step may be a result of improved snow cover. Systematic improvements in the precipitation biases were

not observed. Instead, precipitation biases generally increased with finer horizontal resolution or shorter time step, suggesting
that some tuning may be required in the physics parameters when changing horizontal resolution and time step.

We stress that the results presented in this study are specific to the OpenIFS atmosphere model and are crucial for the modeling
community that uses the OpenIFS in their climate models such as EC-Earth (**Haarsma et al., 2020; Döscher et al., 2022**),
CNRM (**Voldoire et al., 2019**), AWI (**Streffing et al., 2022**), and GEOMAR (**Kjellsson et al., 2020**). However, the results
may also have implications for other climate modeling communities, at least for those that use a semi-Lagrangian scheme
similar to the IFS (**e.g.,Walters et al., 2019**) in the atmospheric component where long time steps are both possible and often
desirable to reduce the computational cost of the model.

The zonal wind bias improvement in the OpenIFS is important for research questions linked with the Southern Ocean dynamics
that plays a crucial role in both the global atmosphere and ocean circulation. We propose that the model time step not be longer
than 30 minutes at any horizontal resolution to minimize surface wind biases over the ocean. The computational cost increases
linearly with dt (time step), whereas the cost scales with horizontal resolution as dx^3 as the number of grid points increases
in both dimensions and the time step is likely shortened as well. Hence, reducing the model time step from 45 or 60 minutes
to 20 or 30 minutes may double the computational cost, but lead to significant improvements in the simulated climate. The
optimal model time step for the OpenIFS coarse resolution model (1°) is suggested to be 30-minute, but should likely be
somewhat shorter, e.g., 15 min, for higher resolutions. In this study, we have not investigated sensitivity of extreme events to
the model time step as our focus is mostly on mean state biases. The effect of model horizontal resolution and time step on
precipitation extremes is the topic of another manuscript currently in preparation.

Another limitation of this study is that the time step sensitivity was only tested for the low-resolution configuration, OIFS-
LRA, and not the higher resolutions, e.g., OIFS-HRA. We found that much of the surface wind biases were alleviated by a
shorter time step due to increased shallow and mid-level convection (**Fig. 3**). We therefore speculate that a similar sensitivity
should be present at high horizontal resolution (~25 km), i.e., a simulation with OIFS-HRA using a 1h time step would most
likely exhibit a much larger surface wind biases than the OIFS-HRA simulation with 15min time step

**Code and data availability**
The OpenIFS model requires a software license agreement with ECMWF, and OpenIFS' license is easily given free of charge
to any academic or research institute. The details of the different versions of the OpenIFS model, including the OpenIFS
version used in this study, i.e., 43R3, can be found at https://confluence.ecmwf.int/display/OIFS/About+OpenIFS. The
OpenIFS model source code has been made available for the editor and reviewers.
The input datasets (both initial and boundary conditions) needed to run the OpenIFS model, run scripts, the model output, and
the Jupiter notebook that support the finding of this study are available at **(Savita, 2023)**. The source code for XIOS 2.5,
revision 1910, is available from the official repository at https://forge.ipsl.jussieu.fr/ioserver/ under CeCILL_V2 license.
OpenIFS experiments were made using ESM-Tools (https://github.com/esm-tools/esm_tools/). The OASIS coupler is
available at https://oasis.cerfacs.fr/en/. The XIOS, ESM-Tools and OASIS coupler used in this study can be downloaded from
https://doi.org/10.5281/zenodo.8189718.
The observational datasets used to validate OpenIFS model results in this study are downloaded from the ERA5
(https://cds.climate.copernicus.eu/), GPCP (https://psl.noaa.gov/data/gridded/data.gpcp.html) and CRUTEM4
(http://badc.nerc.ac.uk/data/cru/) websites. Total model output exceeds 10 Tb and it not publicly available, but is available
from the authors upon reasonable requests.
**Author contributions**
All the model simulations were conducted by AS and JK. Analysis of the output and the writing of text for this paper
coordinated by Savita with substantial contributions from JK, RPK, ML, TR, SW and WP.

**Competing interest**

The authors declare that they have no conflict of interest.

**Acknowledgement**

AS, JK and ML are supported by JPI Climate/Ocean (ROADMAP project grant 01LP2002C). WP was supported by IBS (IBS-
R028-D1). We wish to thank the OpenIFS team at ECMWF for the technical support. All simulations were performed on the
HLRN machine under shk00018 project resources. All analyses were performed on computer clusters at GEOMAR and Kiel
University Computing Center (NESH). We thanks to Anton Beljaars for discussion on ECMWF model physics. We also thank
both reviewers and the editor for their constructive comments and suggestions during review process.

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

**Table**

| Experiment Name | Horizontal resolution | Vertical grid | Time step | CHPSY | SYPD |
|---|---|---|---|---|---|
| **OIFS-LRA-15m** | Tco95/100km | L91 | 15m | 3.3 k | 11 |
| **OIFS-MRA-15m** | Tco199/50km | L91 | 15m | 13.3 k | 4 |
| **OIFS-HRA-15m** | Tco399/25km | L91 | 15m | 19.2 k | 2 |
| **OIFS-LRA-30m** | Tco95/100km | L91 | 30m | 845 | 21 |
| **OIFS-LRA-1h** | Tco95/100km | L91 | 1h | 256 | 36 |


**Table 1**. List of the experiments performed across different horizontal resolutions and model time steps using OIFS model.
In the above table CHPSY is core hours per simulation year, and SYPD is simulation year per day.



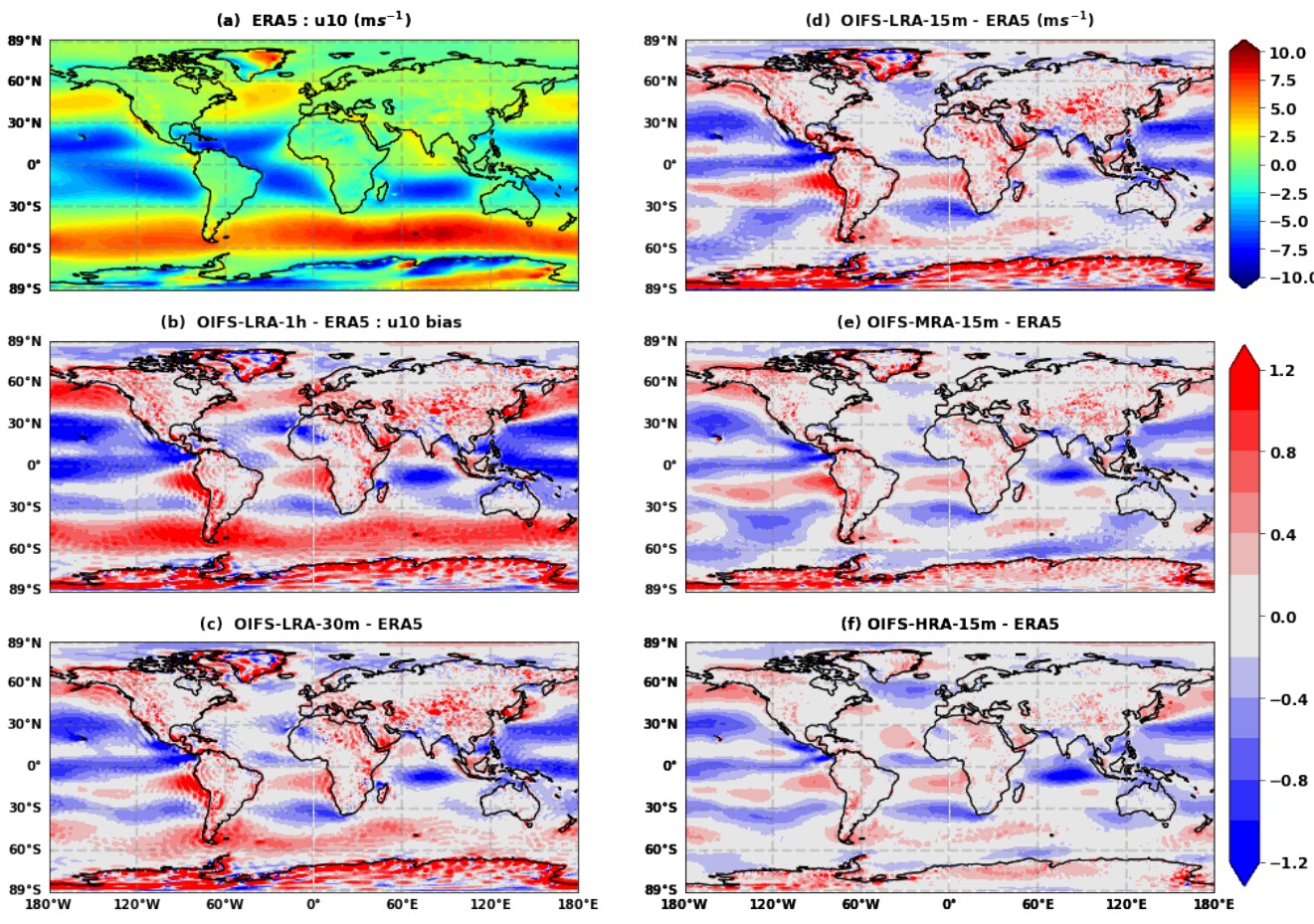


**Figure 1**. (a) Annual mean ERA5 surface zonal wind [ms⁻¹]. (b-d) Annual mean zonal wind [ms⁻¹] bias for different model
time steps (1h (b), 30m (c), and 15m (d)) using ~100 km resolution, and (e-f) with different horizontal resolutions, ~50 (e) and
~25 km (f), respectively. Biases are computed with respect to ERA5 over the period 1979–2019.


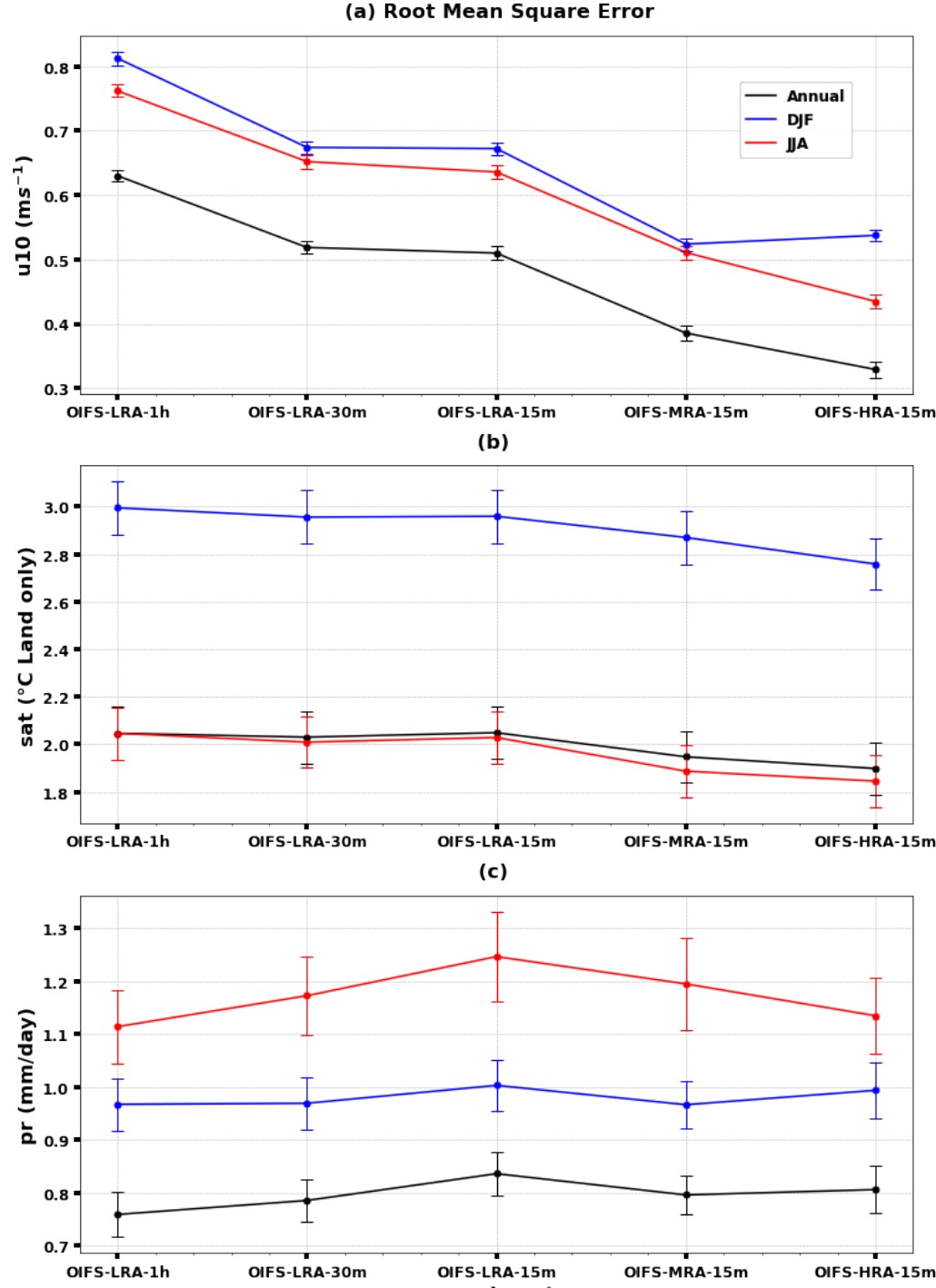


**Figure 2**. Root mean squared error of surface zonal wind (a), SAT (b), and precipitation (c) over the period 1979-2019 for all
the configurations: annual (black) and seasonal mean (DJF: blue, JJA: red). The error bars represent a 95% confidence interval.

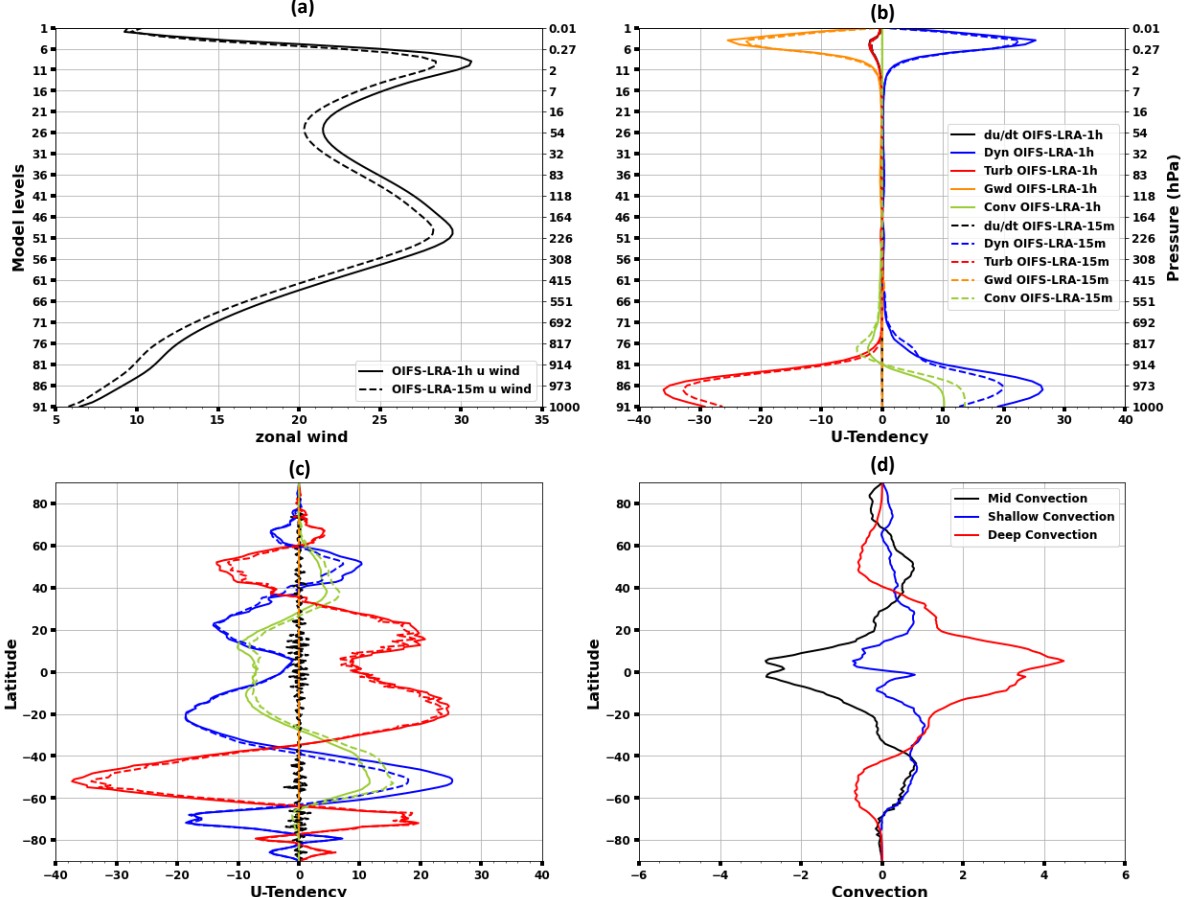



**Figure 3**. (a) Averaged zonal wind (u) [ms⁻¹] and (b) zonal wind tendencies [ms⁻²/hour] over the Southern Ocean (40°S – 60°S,
all longitude) as a function of height for OIFS-LRA-1h and OIFS-LRA-15m. Model levels (y-axis left) and pressure levels (y-
axis right). (c) Zonal and time average of zonal wind tendencies at the lowest level of the model as a function of latitude. (d)
Zonal and time average convection difference [Kgm⁻²/hour] between OIFS-LRA -15m and OIFS-LRA-1h configurations. The
solid lines in panels (b) and (c) show the wind tendency for OIFS-LRA-1h configuration whereas the dashed lines are for
OIFS-LRA-15m configuration. Shown are averages over 1979-2019.

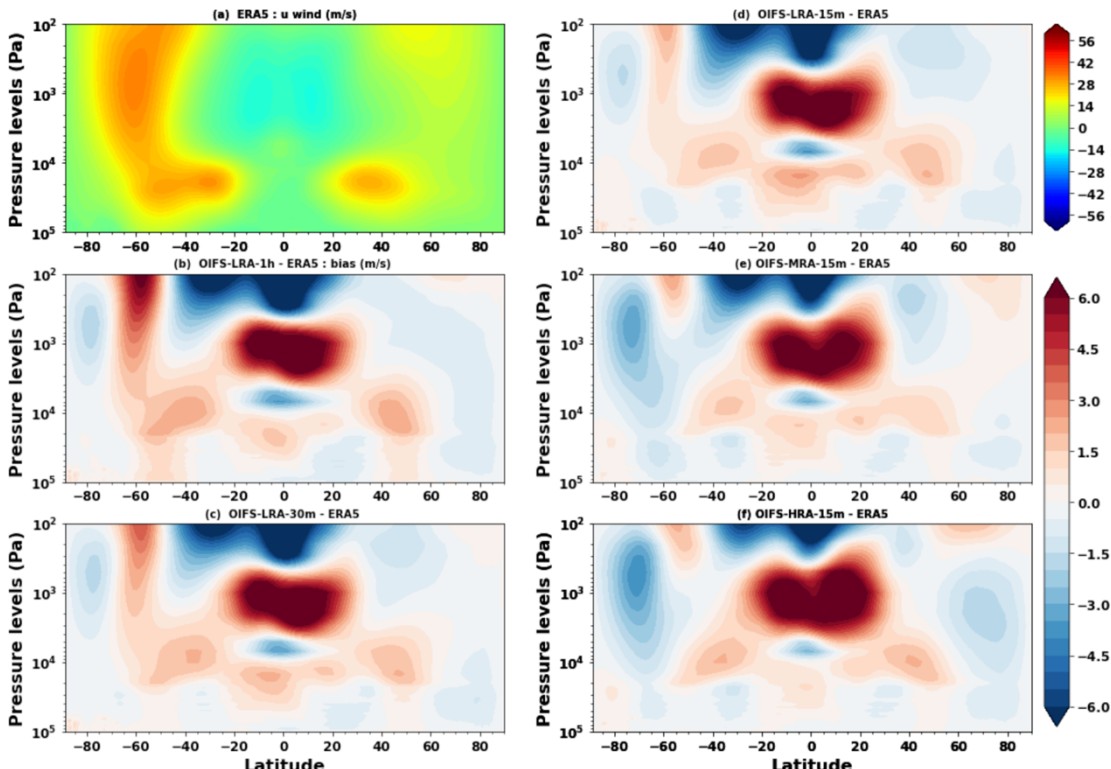

**Figure 4**. (a) Annual zonal mean ERA5 zonal wind [ms⁻¹]. (b-d) Annual zonal mean zonal wind [ms⁻¹] bias for different model time steps (1h (b), 30m (c), and 15m (d)) using ~100 km resolution, and (e-f) with different horizontal resolutions, ~50 (e) and ~25 km (f), respectively. Biases are computed with respect to ERA5 over the period 1979–2019.

633

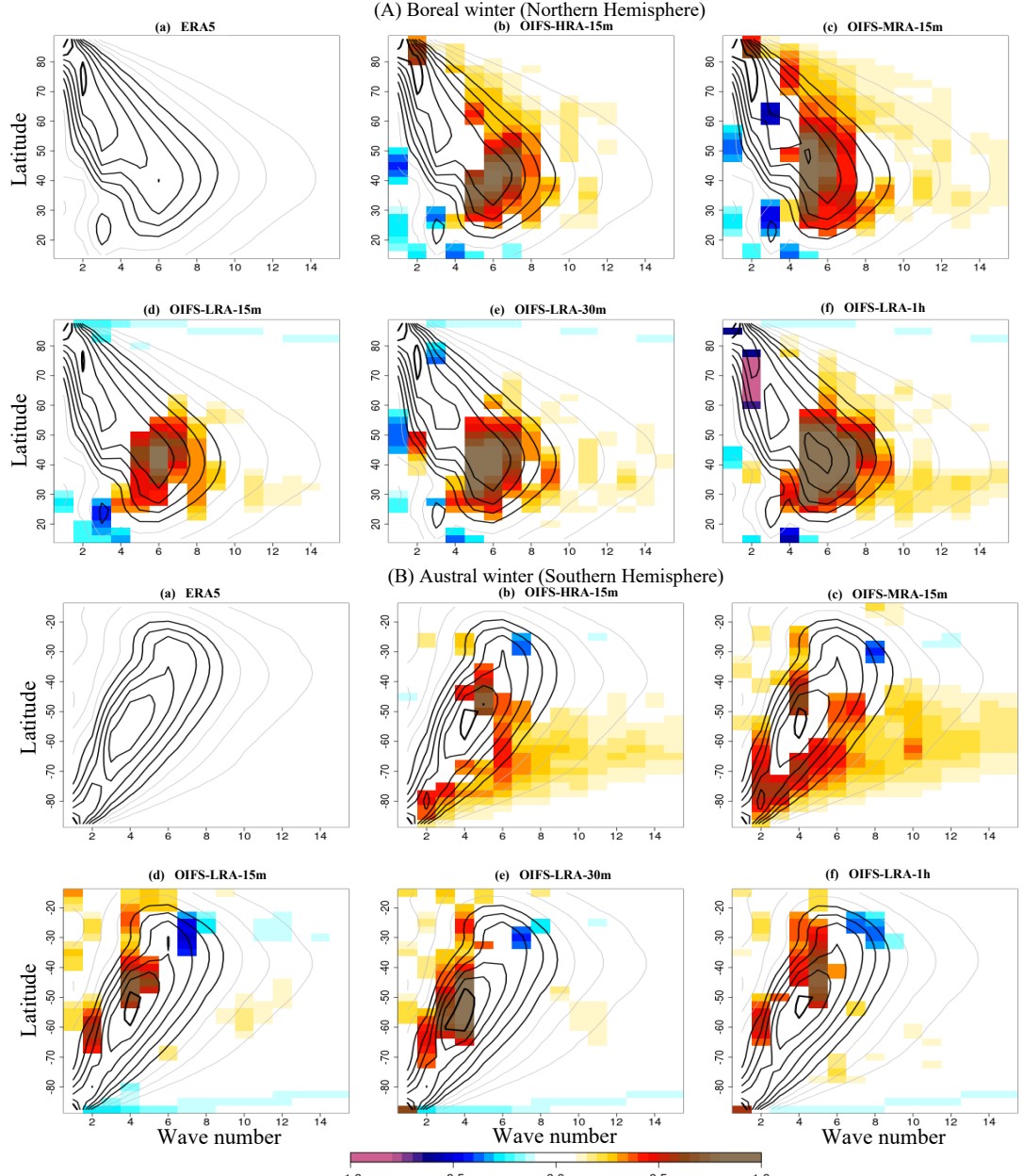

**Figure 5**. (A) The Rossby wave amplitude (contours) for different wave numbers in the Northern Hemisphere at 300 hPa (a) in ERA5 observation and (b-f) in the OIFS model simulations during 1979-2019 in DJF (i.e., boreal winter). The color shows difference of wave amplitude between the model and ERA5 where it is significant on the 95 % confidence level. The wave amplitude and contour interval are shown in ms$^{-1}$. The grey contours start from 2 ms$^{-1}$ and the black contours from 5 ms$^{-1}$ and the contour interval is 1 ms$^{-1}$. (B) is similar to (A), but for JJA (i.e., austral winter).

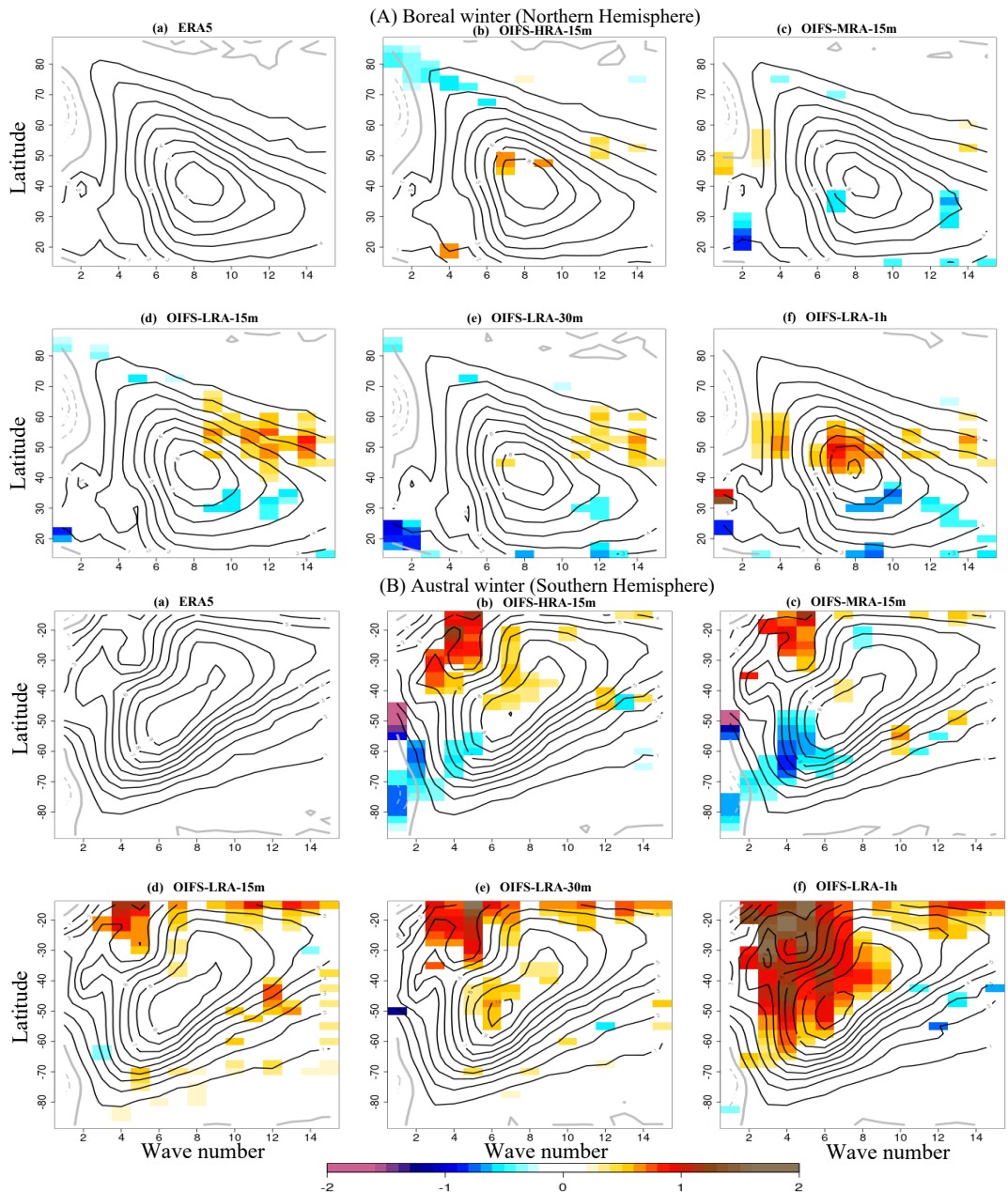

639

**Figure 6.** (A) The Rossby wave phase speed (contours) for different wave numbers at 300 hPa in the Northern Hemisphere in ERA5 (a) observation and (b-f) in the OIFS model simulations during 1979-2019 in DJF (i.e., boreal winter). The color shows the difference of wave phase speed between model and ERA5 where it is significant on the 95 % confidence level. The wave phase speed and contour interval are shown in ms$^{-1}$. The black contours start from 1 ms$^{-1}$and the contour interval is 1 ms$^{-1}$. Panel (B) is similar to panel (A), but for JJA (i.e., austral winter). The dash contours show a negative phase speed and a gray contour shows a zero-phase speed.

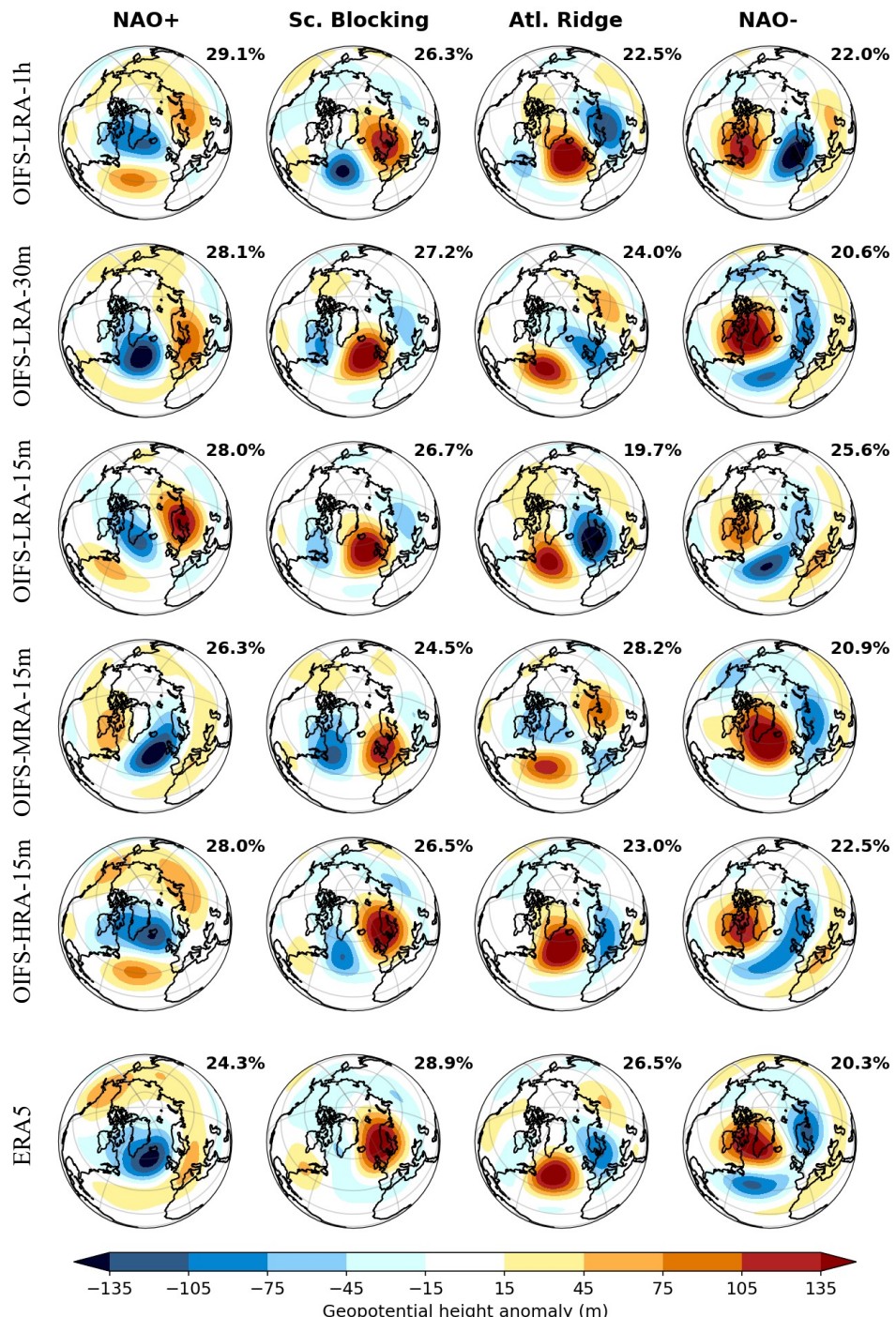

**Figure. 7.** Weather regime patterns over the Euro–Atlantic regions from ERA5 observation (bottom row) and the individual OIFS model simulations (1st to 5th row) over the time period 1979–2019 for DJF (boreal winter season).


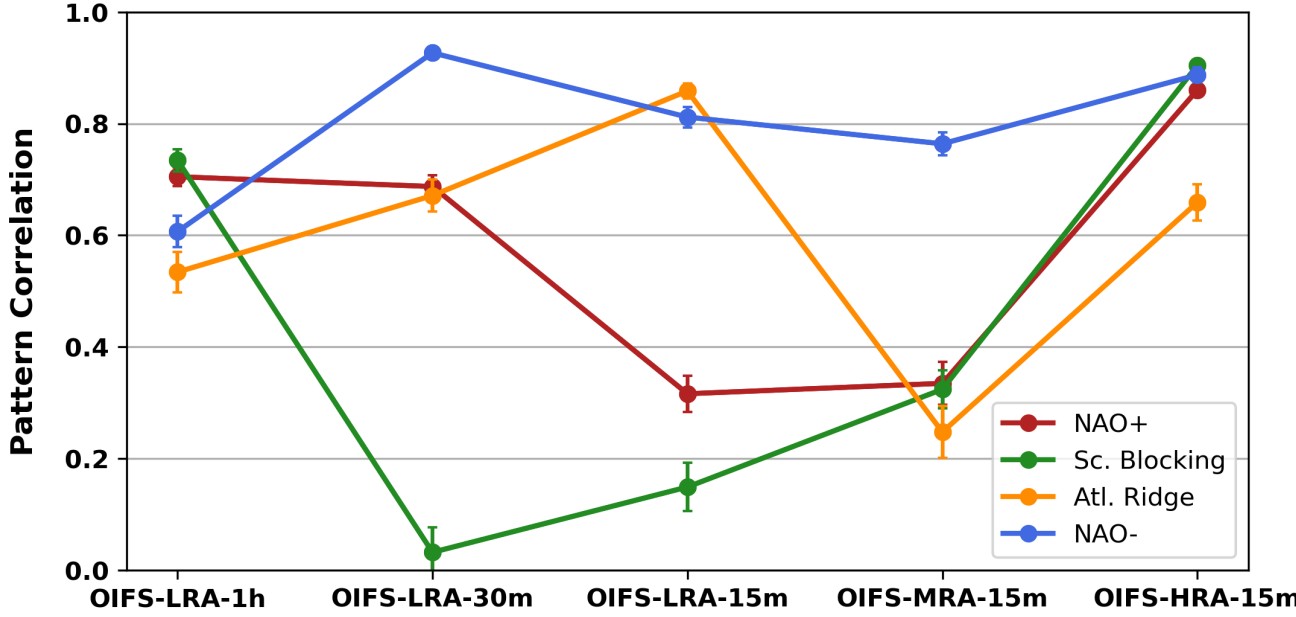


**Figure. 8.** Pattern correlation coefficient of the individual weather regime between OIFS model configurations and ERA5 for
the period 1979-2019 for the DJF season. The error bars represent a 95% confidence interval.



