# Peer review of "Assessment of Climate Biases in OpenIFS Version 43R3 across Model Horizontal Resolutions and Time Steps"

_Geoscientific Model Development, 2023_

## Author Comment (AC1)

**Assessment of Climate Biases in OpenIFS Version 43R3 across Model Horizontal Resolutions and Time Steps**

Abhishek Savita[1], Joakim Kjellsson[1,2], Robin Pilch Kedzierski[3,1], Mojib Latif[1,2], Tabea Rahm[1,2], Sebastian Wahl[1] and Wonsun Park[4,5]

**Referee #1**

Review of "Assessment of Climate Biases in OpenIFS Version 43R3 across Model Horizontal Resolutions and Time Steps"
This study evaluates the sensitivity of an atmospheric general circulation model (OpenIFS cycle 43R3) to different combinations of time-step and horizontal resolution. The authors evaluate several aspects of the mean climate and variability in simulations with prescribed sea surface temperatures (SSTs) and sea-ice over the the period 1979-2019. The authors identify several regions where reducing model time-step from 60 minutes to 15 minutes can have positive impacts on systematic biases that are comparable to the impact of increasing resolution. The manuscript is clear and concise and the topic is within the scope of GMD. The results of this study will likely be of interest to the many users of the IFS model in weather and climate sciences. In particular, these results raise interesting questions about model development strategy when it is often necessary to work with cheaper and/or reduced resolution surrogates of more expensive operational/production configurations. I believe this manuscript can be suitable for publication in GMD but I have several comments that I think would improve and clarify the current manuscript.

Thank you very much, and happy to hear that reviewer finds this manuscript interesting and relevant to IFS modelling community.

**Major comments:**

(1) In section 3.1 the authors focus on biases of near-surface fields, and how these are alleviated with reduced time-step and/or increased resolution.
I encourage the authors to extend their analysis to other levels in the atmosphere (e.g. zonal means of temperature/wind against model/pressure levels). Given the changes in convection and vertical mixing identified later in the paper, I think it is possible that the authors will find similar sensitivities in the troposphere. I also think it is possible that changes at other levels may result in increased rather than reduced biases. This is fine, as the most interesting aspect is the sensitivity to time-step and how this varies with region (e.g. is it limited to near-surface/troposphere). I think it is unlikely that reducing time-step will improve biases in all regions/levels, so it would also be interesting to discuss and interpret any regions of increased bias (e.g. whether they might indicate a role for compensating errors).

We have added an additional section addressing the zonal mean wind and temperature biases (lines 271-296):
"We examined the zonal mean u wind bias at different model levels, and it is shown in Figures 4. We find that zonal mean u wind bias over the tropical region (40°S and 40°N) is positive and independent of model horizontal resolution and model time step (Fig. 4b-f). The OIFS-HRA-15m configuration has a relatively large negative bias in the Northern Hemisphere compared to the other configurations. The OIFS-LRA-15m and OIFS-HRA-15m zonal mean u wind bias is similar to that in Robert et al. (2018). However, the zonal mean u wind bias in the Southern Hemisphere is not consistent across horizontal resolution or model time step. The zonal mean u wind bias in midlatitudes (i.e., 70°S to 50°S) is positive and large in the OIFS-LRA-1h configuration and reduces throughout the pressure levels by shortening the model time step in the coarse resolution OpenIFS configuration (i.e., OIFS-LRA-30m and OIFS-LRA-15m). Whereas, the negative zonal wind bias south 70°S in the coarse resolution configuration is consistent across the different time steps (Fig. 3b-d). It is also interesting to note that both OIFS-MRA-15m and OIFS-HRA-15m configurations exhibit a negative bias over the Southern Ocean (SO) at most of the pressure levels, which is not seen in the standard OIFS-LRA-1h, nor in the OIFS-LRA-30m or OIFS-LRA-15m configurations. Overall, we conclude that by reducing the model time step in the coarse resolution configuration, we improve winds not only at the surface but also at higher model levels mostly over the SO. A similar conclusion does not hold for the OIFS-MRA-15m and OIFS-HRA-15m configurations, as both suffer from large negative bias over the SO.

We also examined the zonal mean temperature bias at different pressure levels. We find a cold bias (1.5 to 6 °C) in the troposphere and lower stratosphere, and a warm bias (1.5 to 6 °C) above the stratosphere across the configurations (Figure not shown). This indicates that OpenIFS simulations (independent of model time step and horizontal resolution) are colder than observations in the lower stratosphere and warmer above. The cold bias in the lower stratosphere is larger in the high resolution (i.e., OIFS-HRA-15m), and the warm bias above the
stratosphere is smaller compared to the other configurations. Robert et al. (2018) noticed a similar zonal mean
temperature bias and speculated that the zonal mean temperature bias is linked with the sensitivity of spurious
mixing due to convection and diffusion."

[Figure]

**Figure 4**. (a) Annual zonal mean ERA5 zonal wind [ms$^{-1}$]. (b-d) Annual zonal mean zonal wind [ms$^{-1}$] bias for
different model time steps (1h (b), 30m (c), and 15m (d)) using ~100 km resolution, and (e-f) with different
horizontal resolutions, ~50 (e) and ~25 km (f), respectively. Biases are computed with respect to ERA5 over the
period 1979–2019.
(2) The abstract concludes with the general statement that "reducing the time step in the OpenIFS model, one
can alleviate some climate biases at a lower cost than by increasing the horizontal resolution." I would like the
authors to to add some discussion of whether they expect their results to generalise to resolutions and/or time-
steps not tested in this manuscript. For example, how far is the LR configuration from converging? Would
reducing time-step in a much higher resolution model (e.g. 9km) bring similar benefits? Depending on these
additions, the authors may wish to qualify the concluding line of the abstract.
As suggested by the reviewer, we have modified the general conclusion to more specific as (lines 28-29):
"Reducing the time step in the coarse resolution (~100 km) OpenIFS model, one can alleviate some climate
biases at a lower cost than by increasing the horizontal resolution.
(3) What is the impact time-step/resolution on the representation of extremes? It is plausible that changes in
time-step that improve the mean state have a limited impact on extremes that are more sensitive to horizontal
resolution (e.g. orographic precipitation or tropical cyclones). As cited by the authors, the mean climate of the
25km and 50 km HighResMIP configurations of IFS are very similar. However, the differences in horizontal
resolution are evident in the representation of extremes (examples below):
https://agupubs.onlinelibrary.wiley.com/doi/full/10.1029/2019JD032184
Bador et al. (2020)
https://journals.ametsoc.org/view/journals/clim/33/7/jcli-d-19-0639.1.xml
Roberts J. et al (2020)
This is an excellent point. However, the main focus in this study is the mean state biases. The OpenIFS' model
time-step sensitivity to extremes will be discussed in detail in a separate manuscript.
**Minor comments:**

Introduction: This section would benefit from an overview of the "expected" impacts of reducing time-step in
simple models in terms of truncation error and how this might not always hold true in a more complex system.
For example, in simple finite difference models, solutions converge as grid-spacing and time-step are decreased
due to reduced truncation errors. The choice of time-step and grid-spacing may also be constrained by stability
criteria. However, this intuition does not always hold in complex models due to the coupling between many
different elements. For instance, it is plausible that the unconditionally stable semi-implicit semi-Lagrangian
scheme used in the IFS allows a user to configure the model with a long time step to reduce the cost. Later
developments on top of this configuration introduce compensating errors in other aspects of the physics that
reduce biases. Reducing the time step at a later stage may then leads to increased biases as the model
configuration has been implicitly tuned for a particular combination of time-step and resolution.

We added some text to the Introduction section (see below) (lines 75-82):

"While the semi-implicit semi-Lagrangian scheme, as used in OpenIFS, is unconditionally stable and the time
step can be chosen to be very long, a shorter time step generally leads to a decrease in truncation error in the
finite differences and thus a more accurate representation of the model dynamics. The physics
parameterisations, which are computed independently of each other in OpenIFS, also benefit from a shorter time
step as it will allow the various parameterisations to be coupled at a higher frequency (Beljaars et al. 2018).
However, model parameters for e.g., convection or diffusion may be tuned for a specific time step and
shortening the time step can therefore, in some cases, increase model error. Hence, a shorter model time step is
expected to reduce biases in model dynamics, e.g., winds, while the results for parameterised processes, e.g.,
precipitation, may be mixed".
Lines 110-119. The authors state that their "detailed analysis of phase speed in such a manner is novel in
literature". This may be the case, but I would like the authors to provide a more detailed summary of the method
used to diagnose the amplitude and phase speed of extratropical Rossby waves (e.g. a bullet point list of the
main processing steps). The current description is insufficient for reproduction of the analysis. In particular, it is
not clear from the text how wave packets and associated phase speeds are diagnosed.
We revised the texts so that we can reproduce the analysis by following the steps (lines 132-137)
"We then applied the Fourier decomposition analysis to determine amplitude and position for each Rossby wave
number at each latitude as a function of time. Phase speed is computed as the difference in the daily position of
each wave, and stored at the midpoints in the time dimension. For consistency, wave amplitudes are interpolated
to the midpoints in time as well. Lastly, seasonal averages are computed from the daily data for the boreal and
austral winter seasons over the time period 1979–2019.  In the case of phase speed, it is weighed by the
corresponding daily (midpoint) amplitude squared when computing the seasonal averages in order to account for
the impact of higher-amplitude events."
Line 169. Typo? Should be "Roberts et al. (2018)" as in intro?
This has been fixed.
Line 198. Is it correct to include Coriolis? Work done by Coriolis term should be zero since it acts perpendicular
to motion of air parcels.
Yes, the Coriolis term is included in the DYN part because we are analyzing zonal wind tendency. However, we
have not quantified the individual contribution of the Coriolis term to the DYN term.
Lines 226-228: It is possible that the lower precipitation RMSE in OIFS-LRA-1h is due to a "double penalty"
effect that penalises higher resolution models, which have more structure in the precipitation fields. Is the
precipitation in the LR-1h experiment notably smoother? Other metrics (e.g. fractions skill score) may provide a
different ranking of models. More details on double-penalty effects and fraction skill score here:
https://www.ecmwf.int/en/about/media-centre/science-blog/2023/verifying-high-resolution-forecasts
To verify our result, we computed the RMSE by smoothing the data and found a similar conclusion. However,
the RMSE values differ in magnitude if we compare them with without smoothing. We have not computed the
fraction skill score as our conclusions are insensitive to the double penalty effect (see the figure below; (left)
RMSE and (right) time mean Precipitation). We now added this information in the main text as well (lines 259-
263).
"We have computed the SAT and precipitation biases with a 3-point smoothing, i.e., approximately 3x3 degree
spatial smoothing, which eliminates the wiggles near steep topography arising from the Gibbs' phenomenon in
the model spectral fields. We find that smoothing the fields does not change the main result that precipitation biases increase with shorter time step in Tco95 and then decreases somewhat with higher horizontal resolution.
Hence, the wiggles are not the main source of precipitation biases and their presence does not impact the
findings in this study."

[Figure]

Lines 237-238 and figure 4: Why do the authors standardise the wave amplitude biases in figure 4 instead of
showing the absolute values? This standardisation emphasises errors in regions the authors argue are
unimportant, which complicates interpretation of the plots. Specifically, the  authors focus their analysis of
Rossby waves on the "region where the wave amplitude is larger than 5 ms-1 is termed core region, which
mostly covers the area that is occupied by the thick black contours in Fig. 4". However, biases are presented
"relative to ERA5 (model – ERA5), normalized by the ERA5 detrended  variability expressed by the standard
deviation", which highlights errors in the high-latitude high-wavenumber waves that are dismissed by the
authors as "unimportant as these waves have a small amplitude and little effect on variability".
We updated the figures with absolute error and modified the texts accordingly (pages 9-10 and also provided
texts below).

[Figure]

Line numbers:

[revised manuscript text omitted]

Section 3.2. How do the authors interpret the impact on Rossby wave amplitude/phase speed biases? For
example, is it related to the representation of tropospheric jets and associated wave guides and their biases?
The representation of tropospheric jets and associated wave guides can be related to biases in Rossby wave
packets ( e.g., Giannakaki and Martius, (2016), Hakim, 2005 and Baumgart et al., 2018). We evaluate the
amplitude and speed of each wavenumber individually that indicating at which scale the model biases occur.
RWPs are then the combination (or sum) of the intermediate ones (e.g. wavenumbers 4-15), but
diagnosing/tracking RWPs is a different analysis, and there's no consensus on the best method (Wolf and Wirth,
2017).
Section 3.3 and figure 7. What is the sampling uncertainty in these composites and estimates of pattern
correlation (e.g. estimated using bootstrap resampling of available dates)? Are the differences between
configurations significant?
We added sampling uncertainty using bootstrapping method with random 2000 iterations.

[Figure]

Figure. 8. Pattern correlation coefficient of the individual weather regime between OIFS model configurations
and ERA5 for the period 1979-2019 for the DJF season. The error bars represent a 95% confidence interval.

Table 1. What is the HPC cost of the different configurations (e.g. core hours per model year)? Do they scale as
expected from changes in time-step and number of grid points?
As the reviewer suggested, we added this information to Table 1 (lines 694-698).
Figure 2. What is the sampling uncertainty in these estimates of RMSE? Are the differences in RMSE between
configurations significant?
We added the figure depicting RMSE with sampling uncertainty using bootstrapping method with random 2000
iterations (see below and page 22).

[Figure]

Figure 2. Root mean squared error of surface zonal wind (a), SAT (b), and precipitation (c) over the period 1979-
2019 for all the configurations: annual (black) and seasonal mean (DJF: blue, JJA: red). The error bars represent
a 95% confidence interval.

---

## Author Comment (AC2)

**Assessment of Climate Biases in OpenIFS Version 43R3 across Model Horizontal Resolutions and Time Steps**

Abhishek Savita[1], Joakim Kjellsson[1,2], Robin Pilch Kedzierski[3,1], Mojib Latif[1,2], Tabea Rahm[1,2], Sebastian Wahl[1] and Wonsun Park[4,5]

**Referee #2**

Overall the manuscript is clear, concise, to the point and relevant. I appreciated the final recommendations on the recommended resolutions and timestep settings. One thing I did miss is a discussion on the wiggles in the surface fields in the LRA and MRA configurations.

This is in fact a known issue that one of the co-authors complained about in the OpenIFS CONFLUENCE page https://confluence.ecmwf.int/pages/viewpage.action?pageId=188034913. In this page it is described that the Tco (octahedral reduced Gaussian grid ) grid is a poor choice for the low resolution configurations, instead I would encourage the use of compatible TL (reduced Gaussian grid) such as TL159 and TL399. Nevertheless, the findings of this study are valid and I do not recommend repeating the exercise using the TL grids, but I would expect some discussion on this in the introduction, why the authors did not use the TL grids for the LRA and MRA configurations, despite the known issues? Also this should be pointed out in the results section.

Thank you very much for raising this concern. Our choice of grid was motivated by finding a usable low resolution for atmosphere-only and coupled experiments that uses little resources. Tco95 was deemed to be the lowest acceptable resolution as the available lower-resolution configurations, e.g., TL95 and Tq42, were too coarse for our interests. As presented in this paper and elsewhere, the lack of spectral filtering introduces spectral wiggles. As shown in this paper, however, these wiggles are not the main source of model biases. For example, RMSE of T2m and precip biases are relatively insensitive to spatial smoothing. We have now added some motivation for our grid choices in sec 2 (lines 108 to 114).

"Our study is partly motivated by evaluating the suitability of various OpenIFS configurations for coupled climate simulations with FOCI-OpenIFS (Kjellsson et al., 2020) with an atmosphere horizontal resolution higher than that of ECHAM6 Tq63/N48 (~200km) in FOCI (Matthes et al., 2020). Our choices thus fall on three different horizontal resolutions: a low-resolution (Tco95, ~100 km), a medium-resolution (Tco199, ~50 km), and a high resolution (Tco399, ~25 km). The Tco95 grid is the lowest acceptable resolution since the supported lower-resolution grids, e.g., Tl95/N48 and Tq42/F32, are either similar to Tq63 in ECHAM6 or coarser. The Tco399 grid was chosen as an upper limit of what is computationally feasible for AMIP integrations and century-long coupled integrations given our computer resources."

- p. 2 line 38 correct "have been widely used"
 It is fixed now.

- p. 2 line 41 correct "which lead to"
 It is fixed now.

- p. 2 line 60 correct "while increasing from"
 It is fixed now.

- p. 2 line 63 correct "when increasing from"
 It is fixed now.

- p. 2 line 66 correct "IFS models"
 It is fixed now.

- p. 3 line 76 cite Döscher et al (2022) (Döscher et al., 2022)in which the timestep of various resolutions is given, instead of Van Noije et al (2021).
Thank you very much for noticing it, we have fixed it.

- p. 4 line 103, explain which CMIP6 forcings are used ? mole fraction of $CO_2$? authors only specify that aerosol and ozone concentrations are from climatology

We modified the sentence:

"The external forcing is identical to that used in the CMIP6 AMIP simulation except for the aerosol and ozone concentrations"

- p. 4 line 105, which scenario exactly ? SSP5-8.5 ?
It is fix now.

- p. 4 line 111, correct "OIFS simulation datasets"
It is fixed now.

- p. 4 line 130, please confirm that the Pearson's correlation is computed
The missing information is added.

- p. 5 line 156, authors should justify in the intro. why they did not run OIFS-HRA at 1h timestep (i.e. for numerical stability) in order to evaluate if the improvements are due to resolution or timestep
We have not done performed time-step sensitivity experiments using OIFS-HRA configuration due to computer restrictions (very expensive).

- p. 5 line 162, correct "large difference"
It is fixed now.

- p. 6 line 194, black lines are not visible, authors should give their values
We have now provided the net tendencies range in the text.

- p. 7 line 205, correct "lower stratosphere and troposphere"
It is fixed now.

- p. 7 line 215, is there any plausible explanation for this? predominance of ocean surface over continental ones?
The tendency magnitudes are stronger over the Southern Ocean than the Northern Hemisphere due to less rough surface in the Southern Hemisphere.

- p. 7 line 220, The is an abrupt transition from analysis of winds to that of temperature, a proper sentence to indicate this change of focus is needed.
We added a sentence to show the transition from wind to temperature.

- p. 7 line 224, Fig. 2b indicates there is no notable improvement in RMSE from the shortened timestep.
We modified this sentence as:
"Compared to the OIFS-LRA-1h, the SAT RMSE decreases with increased horizontal resolution (OIFS-HRA-15m and OIFS-MRA-15m), and there is no notable improvement when shortened the time step (OIFS-LRA-30m and OIFS-LRA-15m) (Fig. 2b)."  (lines 255-257)

- p. 11 line 345, correct "accounting for most variability"
It is fixed now.

- p. 12 line 369, cite cite Doscher (et al 2022) instead or, or in addition to, Haarsma et al 2020
It is fixed now.

---

## Author Response (AR2)

**Assessment of Climate Biases in OpenIFS Version 43R3 across Model Horizontal Resolutions and Time Steps**

Abhishek Savita[1], Joakim Kjellsson[1,2], Robin Pilch Kedzierski[3,1], Mojib Latif[1,2], Tabea Rahm[1,2], Sebastian Wahl[1] and Wonsun Park[4,5]

**Thank you for your revised manuscript, which answers many of the two Referees' comments. However, regarding remarks of Referee #1, I consider that you satisfactorily answered his first major comment but not his two other major comments.**
**Even if you don't have definite answers or analyses on these two points, please add some text in your manuscript to discuss the issues raised, i.e.**

**1. the generalization of your results to resolutions and/or time-steps not tested in this manuscript, the convergence of the LR configuration and the possible effect of reducing time-step in a much higher resolution model**
We have added some text to lines 452-456 as:
"Another limitation of this study is that the time step sensitivity was only tested for the low-resolution configuration, OIFS-LRA, and not the higher resolutions, e.g., OIFS-HRA. We found that much of the surface wind biases were alleviated by a shorter time step due to increased shallow and mid-level convection (Fig. 3). We therefore speculate that a similar sensitivity should be present at high horizontal resolution (~25 km), i.e., a simulation with OIFS-HRA using a 1h time step would most likely exhibit a much larger surface wind biases than the OIFS-HRA simulation with 15min time step".

**2. the impact of time-step/resolution on the representation of extreme; on this point, you may simply state that this is an interesting area of study that will be addressed in another paper.**
We have added this information in the manuscript to lines 448-450 as:
"In this study, we have not investigated sensitivity of extreme events to the model time step as our focus is mostly on mean state biases. The effect of model horizontal resolution and time step on precipitation extremes is the topic of another manuscript currently in preparation".

**Regarding the comments of the 2nd reviewer:**
**• please add some justification on why you did not run OIFS-HRA at 1h timestep (in Section 2 when you introduce Table 1)**
We have added some text to lines 113-117 as:
We note that exploring the effect of different time steps was only done for the lowest horizontal resolution (Tco95, ~100 km). We did not run similar sensitivity experiments for the high-resolution configuration (Tco399, ~25 km) for two reasons. First, the high-resolution configuration is very computationally expensive. Second, it was deemed more important to explore time step sensitivity at low resolution since this configuration (and other similar resolutions) is often used for coupled climate simulations. The potential time-step sensitivity at high-resolution is discussed in the Discussion section.

**• please consider adding some text with an explanation on why wind tendency magnitudes are stronger over the Southern Ocean than the Northern Hemisphere.**

We have added this information now (lines 239-241):

"The larger magnitudes of the tendencies over the Southern Ocean compared to similar latitudes on the Northern Hemisphere is likely due to the Southern Hemisphere having fewer continents in midlatitudes than the Northern Hemisphere and thus the surface is less rough and allows for stronger winds."